# Discovery and characterization of cross-reactive intrahepatic antibodies in severe alcoholic hepatitis

Ali Reza Ahmadi[1†], Guang Song[2†‡], Tianshun Gao[3†§], Jing Ma[4], Xiaomei Han[3#], Ming-Wen Hu[3], Andrew M Cameron[1], Russell N Wesson[1], Benjamin Philosophe[1], Shane Ottmann[1], Elizabeth King[1], Ahmet Gurakar[5], Le Qi[1], Brandon Peiffer[1], James Burdick[1], Robert Anders[6], Zhanxiang Zhou[7], Hongkun Lu[4], Dechun Feng[4], Chien-Sheng Chen[8], Jiang Qian[3], Bin Gao[4], Heng Zhu[2]*, Zhaoli Sun[1]*

[1]Department of Surgery, Johns Hopkins University School of Medicine, Baltimore, United States; [2]Department of Pharmacology and Molecular Sciences, Johns Hopkins University School of Medicine, Baltimore, United States; [3]Department of Ophthalmology, Johns Hopkins University School of Medicine, Baltimore, United States; [4]Laboratory of Liver Diseases, National Institute on Alcohol Abuse and Alcoholism (NIAAA), National Institutes of Health (NIH), Baltimore, United States; [5]Department of Medicine, Johns Hopkins University School of Medicine, Baltimore, United States; [6]Department of Pathology, Johns Hopkins University School of Medicine, Baltimore, United States; [7]Center for Translational Biomedical Research and Department of Nutrition, University of North Carolina at Greensboro, North Carolina Research Campus, Kannapolis, United States; [8]Department of Food Safety/Hygiene and Risk Management, National Cheng Kung University, Tainan, Taiwan

*For correspondence:
hzhu4@jhmi.edu (HZ);
zsun2@jhmi.edu (ZS)

†These authors contributed equally to this work

Present address: ‡School of Life Sciences, Central China Normal University, Wuhan, China; §Research Center, The Seventh Affiliated Hospital of Sun Yat-sen University, Shenzhen, China; #Office of Translational Science, U.S. Food and Drug Administration, Silver Spring, United States

Competing interest: The authors declare that no competing interests exist.

**Abstract** The pathogenesis of antibodies in severe alcoholic hepatitis (SAH) remains unknown. We analyzed immunoglobulins (Ig) in explanted livers from SAH patients (n=45) undergoing liver transplantation and tissues from corresponding healthy donors (HD, n=10) and found massive deposition of IgG and IgA isotype antibodies associated with complement fragment C3d and C4d staining in ballooned hepatocytes in SAH livers. Ig extracted from SAH livers, but not patient serum exhibited hepatocyte killing efficacy. Employing human and *Escherichia coli* K12 proteome arrays, we profiled the antibodies extracted from explanted SAH, livers with other diseases, and HD livers. Compared with their counterparts extracted from livers with other diseases and HD, antibodies of IgG and IgA isotypes were highly accumulated in SAH and recognized a unique set of human proteins and *E. coli* antigens. Further, both Ig- and *E. coli*-captured Ig from SAH livers recognized common autoantigens enriched in several cellular components including cytosol and cytoplasm (IgG and IgA), nucleus, mitochondrion, and focal adhesion (IgG). Except IgM from primary biliary cholangitis livers, no common autoantigen was recognized by Ig- and *E. coli*-captured Ig from livers with other diseases. These findings demonstrate the presence of cross-reacting anti-bacterial IgG and IgA autoantibodies in SAH livers.

## eLife assessment

This **important** study tested the hypothesis that liver-derived but not serum-derived antibodies that are cross-reactive to E.coli and to host proteins can play a role in the hepatic damage found in severe alcoholic hepatitis (SAH). Using a **solid** methodology that includes state-of-the-art microscopy, proteome arrays, and gene ontology assays, it provides strong evidence that liver-derived IgG

and IgA with cytotoxic properties and reactivity to both gut-derived *E. coli* and autoantigens accumulated in hepatocytes of SAH patients but not of healthy controls. The study would benefit from a broader analysis of gut microbiota proteome and further characterization of B cells infiltrating the liver tissue including their numbers/field and their origin (infiltrating versus resident cells). The work opens new avenues of understanding for the pathogenesis of severe alcoholic hepatitis and is of great interest to researchers and clinicians in the field.

## Introduction

Severe alcoholic hepatitis (SAH) is a distinct clinical syndrome that can develop suddenly and quickly lead to liver failure. It carries a particularly poor prognosis with a 28-day mortality ranging from 30% to 50% (*Dugum et al., 2015*; *Thursz and Morgan, 2016*; *Sehrawat et al., 2020*). Unfortunately, there is little to offer medically to such critically ill patients beyond supportive care with steroids, which improves survival in only a minority.

Studies have established connections between alcohol abuse, disruption of gut microbial homeostasis, and alcoholic liver disease (ALD). It has been speculated for over four decades that antibodies targeting intestinal microbes might play a role in pathogenesis of ALD (*Bjorneboe et al., 1972*; *Simjee et al., 1975*; *Kanagasundaram et al., 1977*; *Kater et al., 1979*; *Trevisan et al., 1983*; *Koskinas et al., 1992*). For example, the presence of IgA and IgG on the cell membrane of hepatocytes was detected by direct immunofluorescence in patients with ALD, and the percentage of IgG-positive hepatocytes correlated with transaminase levels, independently of the histological findings (*Trevisan et al., 1983*).

Although a number of studies demonstrated liver IgA deposition in ALD in the 1980s, other reports concluded that IgA deposition in the liver was not specific for ALD but might reflect the reduced metabolism of the damaged livers (*van de Wiel et al., 1987*; *Amano et al., 1988*) or the clearance of excess IgA from the circulation (*Nagura et al., 1989*). A recent study (*Moro-Sibilot et al., 2016*) confirmed that human livers contained IgA-secreting cells originating from Peyer's patches and directed against intestinal antigens. Interestingly, livers from mice with ethanol-induced injury contain increased numbers of IgA-secreting cells and have IgA deposits in sinusoids (*Moro-Sibilot et al., 2016*).

The primary aim of this study was to determine if there was antibody deposition in SAH livers and whether antibodies extracted from SAH livers exhibited hepatocyte killing efficacy. The second aim was to determine if antibodies deposited in the liver were cross-reactive antibodies against both bacterial antigens and human proteins and whether the cross-reactive antibodies were presented uniquely in SAH livers.

## Results

### Immunoglobulins in ballooned hepatocytes in SAH patients

To determine whether antibodies deposit in SAH livers, we collected explanted liver tissues from SAH patients during liver transplantation at Johns Hopkins. Liver tissue sections with H&E staining from SAH patients showed histologic features of SAH including macrovesicular steatosis, neutrophilic lobular inflammation, ballooning hepatocyte degeneration, Mallory-Denk bodies, and portal and pericellular fibrosis (*Figure 1A*). Immunohistochemistry (IHC) staining by using anti-human immunoglobulin (Ig) antibodies demonstrated massive IgA and IgG deposition in ballooned hepatocytes in SAH livers, while none of the hepatocytes were stained with anti-human Ig antibodies in liver tissue sections from healthy donors (HD) except for positive staining in some hepatic sinusoid cells (*Figure 1B and C*). To further confirm the deposition of Ig in SAH livers, the presence of Ig in liver tissue homogenates form SAH (n=7) or HD (n=7) was assessed by western blot analysis and ELISA assays. Western blot analysis demonstrated that the levels of IgA and IgG were dramatically increased in all SAH livers as compared with the donor livers (*Figure 1D and E*). The IgM but not the IgE level was also significantly increased in SAH livers. The increase of IgA, IgG, and IgM levels in SAH liver tissue homogenates was further confirmed by ELISA. IgA and IgG isotypes were major Ig in SAH livers (*Figure 1F*). Further analysis of IgG subclasses demonstrated that the IgG subclass levels – predominantly IgG1 – were significantly higher in SAH livers than that in HD (*Figure 1G*). On the basis of these findings, we performed IHC staining for human IgG in SAH livers from 45 patients with liver transplantation and 10 donor livers in

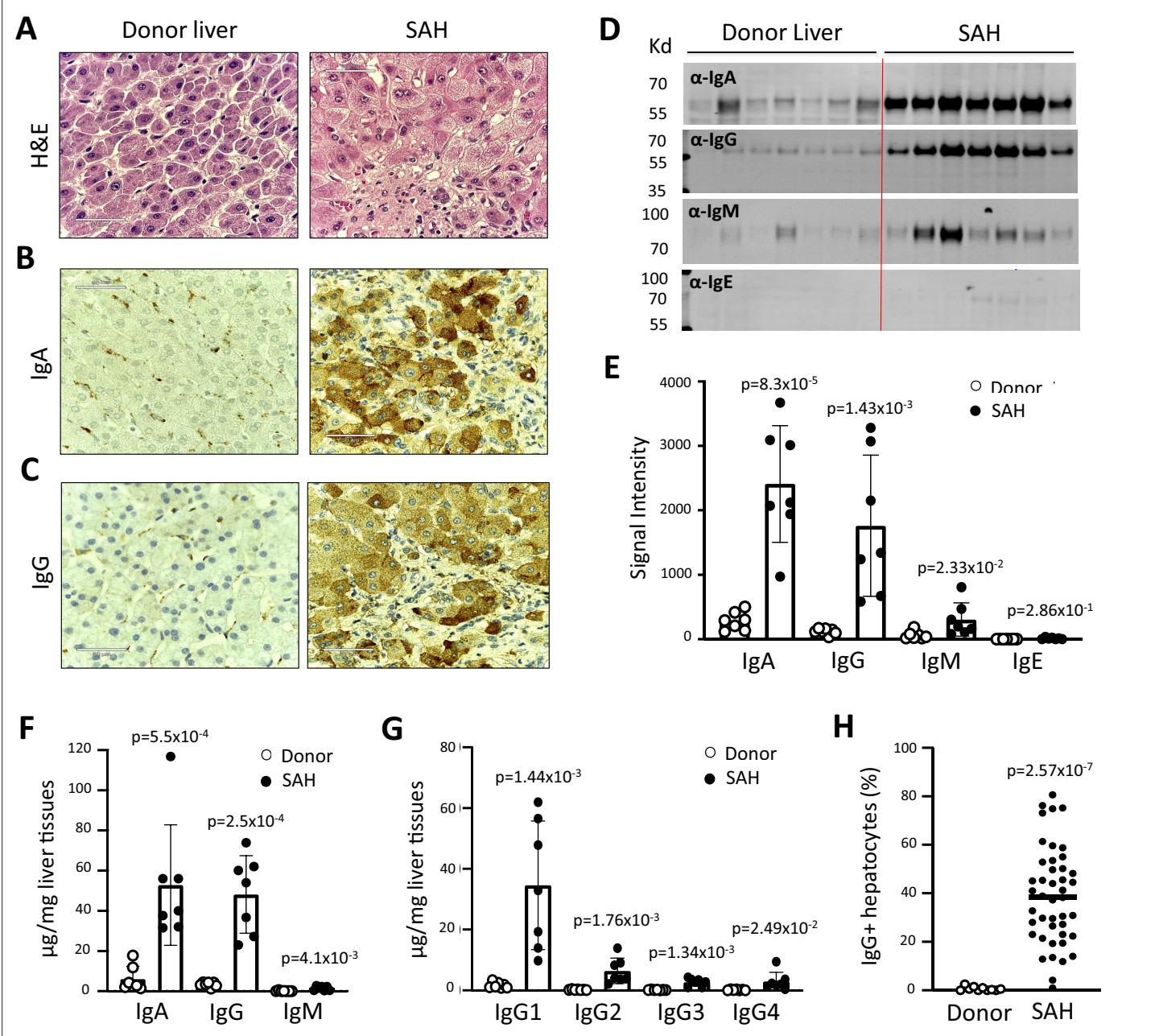

**Figure 1.** Immunoglobulin (Ig) deposition in ballooned hepatocytes of explanted livers from severe alcoholic hepatitis (SAH) patients. (A) Liver tissue sections with H&E staining showed histologic features of SAH. (b, c) Immunohistochemistry staining by using anti-human IgA (B) or IgG (C) antibodies demonstrated IgA and IgG deposition in ballooned hepatocytes in SAH livers. Representative tissue sections from 45 SAH or 10 healthy donor (HD) livers. (D–E) Ig levels in liver tissue homogenates from SAH or HD (n=7/group) were quantified by western blot analysis (D). Western blot analysis demonstrated that the levels of IgA, IgG, and IgM were significantly increased in SAH livers as compared with the HD livers (E). (F–G) Ig isotypes (f) and IgG subclass levels (G) were quantified by ELISA (n=7/group). (H) IgG-positive hepatocytes in tissue sections from 45 SAH patients and 10 HD were quantified by immunohistochemistry staining and using HALO Image Analysis Software.

The online version of this article includes the following figure supplement(s) for figure 1:

**Figure supplement 1.** IgG deposition.

a clinical pathology lab at Johns Hopkins in a double-blind manner. The IgG+ hepatocytes in scanned slides of stained tissues sections were analyzed by using HALO Image Analysis Software. Positive cells were reported as percentage stained surface area of total annotated area by digital analysis (*Figure 1—figure supplement 1*). Few IgG+ hepatocytes were identified in donor livers. In contrast,

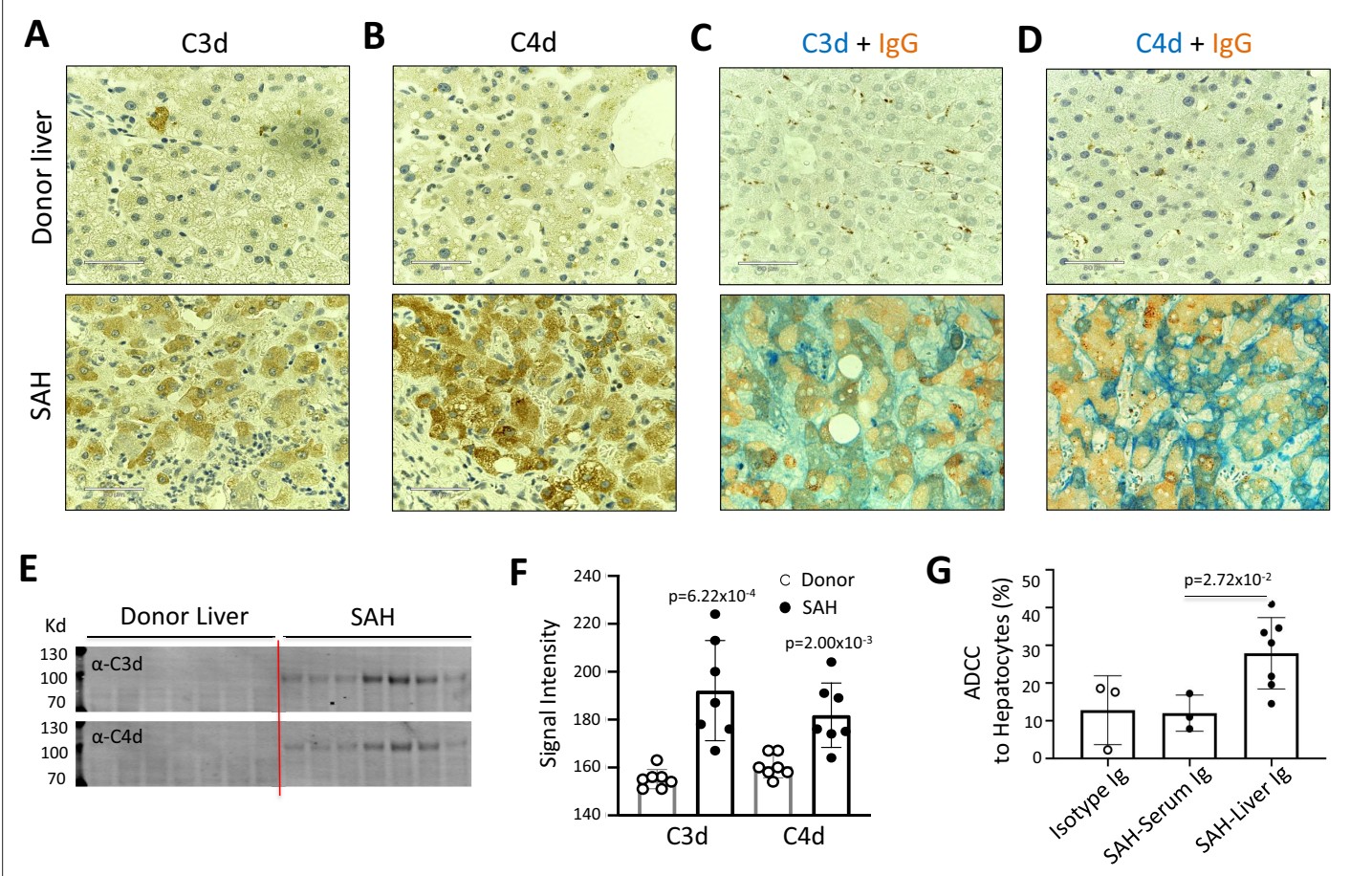

**Figure 2.** Immunoglobulin (Ig) deposition is associated with activation of complement in ballooned hepatocytes and Ig extracted from severe alcoholic hepatitis (SAH) livers exhibits hepatocyte killing efficacy in vitro. To determine if immunoglobulin in ballooning hepatocytes induces activation of complement, complement fragments C3d and C4d were analyzed in SAH livers. (**A–B**) Immunohistochemistry staining showed the presence of both C3d (**A**) and C4d (**B**) in ballooning hepatocytes in SAH livers but not in the donor livers. (**C**) Double staining for IgG and complement fragments C3d or C4d showed IgG co-stained with C3d or C4d in ballooning hepatocytes. Representative tissue sections from seven samples per group. (**E–F**) C3d and C4d levels in SAH livers were quantified by western blot analysis (n=7). (**G**) Ig extracted from SAH livers but not serum exhibit hepatocyte killing efficacy in antibody-dependent cell-mediated cytotoxicity (ADCC) assay. Representative data from three independent experiments.

on average ~40% of the hepatocytes (ranging from 4% to 80%) were IgG positive in the SAH livers (*Figure 1H*). These findings demonstrated the deposition of Ig antibodies in ballooned hepatocytes in SAH livers.

## Ig deposition is associated with activation of complement in ballooned hepatocytes and Ig extracted from SAH livers exhibits hepatocyte killing efficacy in vitro

IgG, especially IgG1, plays a critical role in the classical complement activation pathway. To determine if Ig in ballooning hepatocytes induces activation of complement, C3d and C4d were analyzed in SAH livers. IHC staining showed the presence of both C3d and C4d in ballooning hepatocytes in SAH livers but not in the donor livers (*Figure 2A and B*). Double staining for IgG and complement fragments C3d or C4d showed IgG co-stained with C3d or C4d in ballooning hepatocytes (*Figure 2C and D*). These results indicated that IgG deposition in hepatocytes was associated with activation of complement. Furthermore, complement activation including the presence of C3d and C4d in SAH liver was confirmed by western blot analysis (*Figure 2E and F*). Finally, we asked whether antibodies (Ig) extracted from SAH livers exhibit hepatocyte killing efficacy in an antibody-dependent cell-mediated cytotoxicity (ADCC) assay. Compared to isotype control human Ig from HD, no increased hepatocyte killing was observed when peripheral blood mononuclear cells (PBMCs) (effector cells) from HD

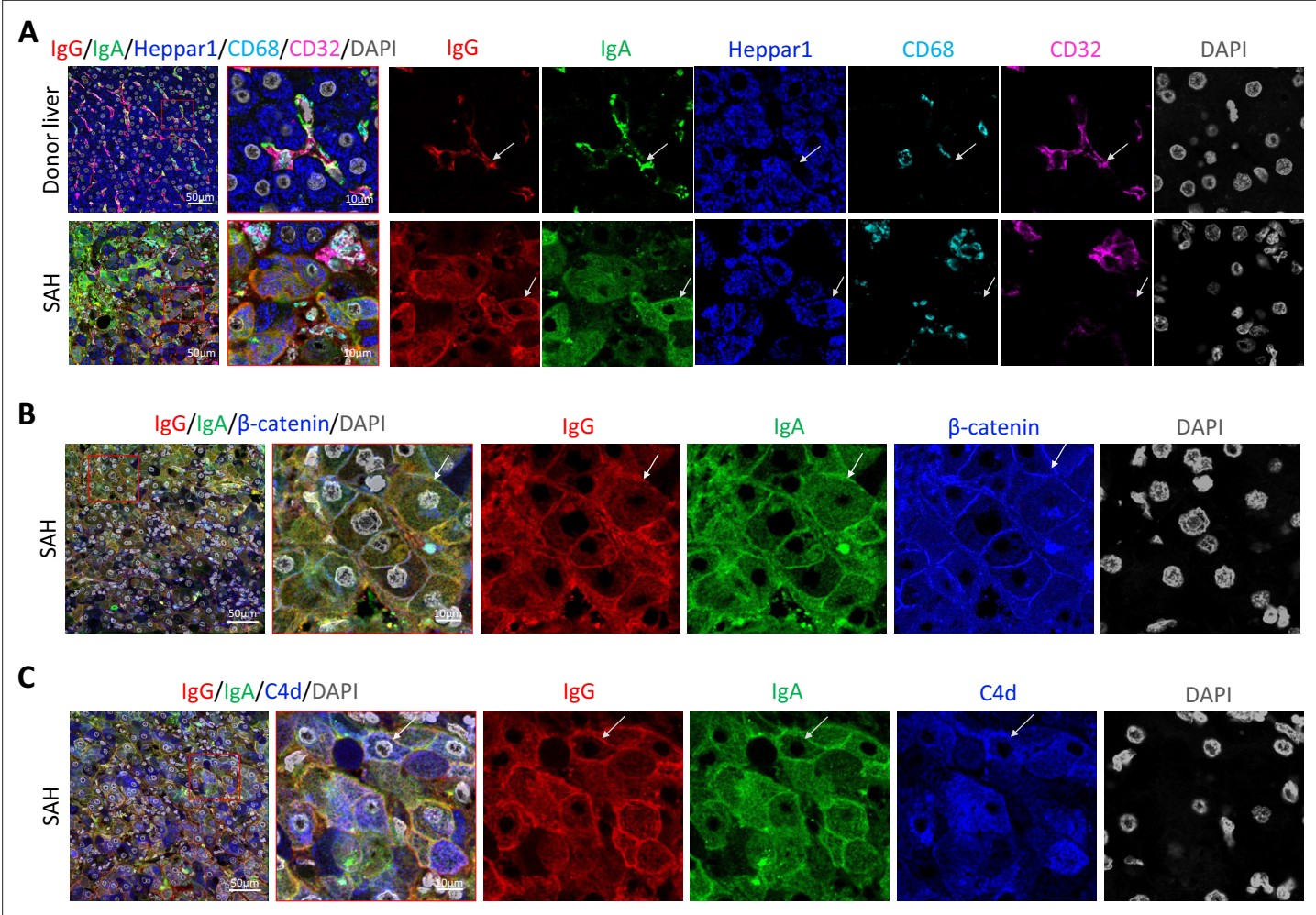

**Figure 3.** To determine if immunoglobulin (Ig) recognize cell surface antigens of ballooned hepatocytes, multiplex immunofluorescence staining was performed in liver tissue sections. (**A**) Images of confocal microscopy showed the presence of both IgG (red) and IgA (green) in ballooning hepatocytes (blue) in severe alcoholic hepatitis (SAH) livers (lower panels), while only hepatic sinusoid endothelial cells (CD32+, purple) stained with IgG and IgA in donor livers (upper panels). (**B**) Co-staining with β-catenin (blue) demonstrated IgG (red) and IgA (green) deposition on membrane of ballooning hepatocytes in SAH livers. (**C**) Triple staining for IgG (red), IgA (green), and C4d (blue) showed both IgG and IgA co-stained with C4d on the surface of hepatocyte. Representative tissue sections from six samples per group.

were added into cultured human hepatocytes (target cells) in the presence of serum Ig from SAH patients. However, the hepatocyte killing efficacy was significantly increased when the same levels of Ig extracted from SAH livers were added into the hepatocytes/PBMCs co-culture system (*Figure 2G*). These results demonstrated that Ig antibodies deposited in hepatocytes of SAH livers could induce activation of complement but more importantly, exhibited antibody-dependent cellular cytotoxicity of hepatocytes. Therefore, deposition of Ig antibodies may contribute to the hepatocyte ballooning degeneration and necrotic damage in SAH.

To further define the Ig and C4d deposition on the membrane of ballooned hepatocytes, we performed multiplex analyses by co-staining liver tissue sections with multiple cell markers for hepatocytes (Heppar1), Kupffer cells (CD68), hepatic sinusoid endothelial cells (CD32), and hepatocyte membrane (β-catenin). Images of confocal microscopy showed the presence of both IgG and IgA in hepatic sinusoid endothelial cells but not hepatocytes in the donor livers, while the majority of ballooned hepatocytes co-stained with IgG and IgA in SAH livers (new *Figure 3A*). Further, co-staining with β-catenin detected heavy IgG and IgA deposition on hepatocyte membrane (*Figure 3B*). Interestingly, the IgG and IgA on hepatocyte membrane were co-stained with C4d as well (*Figure 3C*).

## Human proteome array-identified autoantigens were recognized by Ig extracted from SAH livers

We used the human proteome microarray (HuProt), comprising 21,240 individual purified human proteins, to perform antibody profiling assays (*Hu et al., 2017*). Each liver specimen was treated to release tissue-deposited Ig (*Figure 4A*). After neutralization, the extracted antibodies from each liver sample were separately probed to the HuProt arrays, using isotype-specific secondary antibodies to obtain the IgG, IgA, IgM, and IgE autoimmune signatures of the same liver sample (*Figure 4B*). Many positive human proteins were recognized by each of the four Ig isotypes in all five SAH samples. Each antibody profiling assay was performed in duplicate and only those reproducible signals were scored. A substantial fraction of autoantigens was shared by the antibodies in all five SAH livers, regardless of the Ig isotype (*Figure 4C*, *Figure 4—figure supplement 1*). The total numbers of the shared autoantigens recognized by the IgG, IgA, IgM, and IgE isotypes were 346, 319, 194, and 10 (*Figure 4—figure supplement 1*), respectively, suggesting that the shared autoantibodies of the IgG and IgA isotypes were the most prevalent, while the counterparts of IgE isotype were the scarcest. With tissues of five donor livers, the numbers of positive human proteins were much lower (*Figure 4C* ). Although 95 shared IgG autoantigens were identified by the donor liver samples, 79 (83.2%) of them were also shared by the SAH livers (*Figure 4C*). More importantly, a large fraction (i.e., 267 proteins) of the SAH-shared autoantigens were not found in the donor livers (*Figure 4C*).

We applied the above approach to a group of five livers explanted from patients with AC, AIH, primary biliary cholangitis (PBC), NASH, HCV, and HBV infection. SAH still exhibited the highest number of shared IgG autoantigens, while HBV showed the highest number of shared IgM, IgA and IgE autoantigens, and PBC showed the highest number of shared IgM autoantigens (*Figure 4—figure supplement 2*). The numbers of shared IgG, IgA, and IgM autoantigens were much higher in SAH livers (n=859) as compared with AC (n=349), HBV (n=735), other liver diseases (n<428), or HD livers (n=448). Although the shared IgE autoantigens were the lowest in all seven liver diseases, each disease showed a distinct autoantibody signature (*Figure 4—figure supplement 2*).

We compared the shared autoantigens recognized by SAH Ig to their counterparts from the other six liver diseases. By Venn diagram analysis 45, 41, 68, and 8 autoantigens were commonly recognized by IgG, IgA, IgM, and IgE isotype autoantibodies in liver tissue extracted from different liver diseases (*Figure 4—figure supplement 3*). 188 unique IgG autoantigens, 45 unique IgA autoantigens, and 7 unique IgM autoantigens were recognized by tissue homogenates from the SAH livers, whereas the second highest in this category was HBV livers in which 1 unique IgG autoantigen, 88 unique IgA autoantigens, and 25 unique IgM autoantigens were identified (*Figure 4D*, *Figure 4—figure supplement 3*). The third in this category was PBC in which 7 unique IgG autoantigens, 2 unique IgA autoantigens, and 81 unique IgM autoantigens were identified. The tissue homogenates from HCV livers recognized 32 unique IgG autoantigens, 4 unique IgA autoantigens, and 3 unique IgE autoantigens, while only 4 unique IgA autoantigens were recognized by tissue homogenates from AC livers (*Figure 4—figure supplement 3*), showing disease-distinct autoantibodies in diseased livers regardless of their etiology. A large number of unique autoantigens were recognized by Ig recovered from SAH livers (*Figure 4—source data 1*), indicating Ig (especially IgG) deposited to the SAH livers (*Figure 1*) might play an important role in pathogenesis.

## Ig from SAH or AC livers recognize a unique set of bacterial antigens

Antibodies secreted into the gut mostly target bacteria and bacterial products (*Kato et al., 2014*). Ig deposited in SAH livers might be from the gut, and these Ig may be antibodies targeting intestinal bacterial antigens. To test this, we employed a bacterial proteome array, comprising 4256 purified *E. coli* proteins encoded by a commensal strain K12, to do antibody profiling assays (*Chen et al., 2009*; *Figure 4E*). Using the same liver tissues and approach described above, we obtained the immune signatures of the 40 livers from 7 diseases and 5 HD in duplicate.

Many more bacterial than human antigens were recognized by Ig from alcoholic livers (AC and SAH) compared with HD livers (*Figure 4—figure supplement 4*). The differences in shared bacterial antigens between alcoholic livers and HD livers were more pronounced across all four Ig isotypes. For instance, only one bacterial antigen was commonly recognized by IgG antibodies from HD livers, but 435 shared bacterial antigens (about 10.2% of 4256 purified *E. coli* proteins) were recognized by IgG antibodies in the five SAH livers (*Figure 4F*), and 466 shared bacterial antigens were recognized by

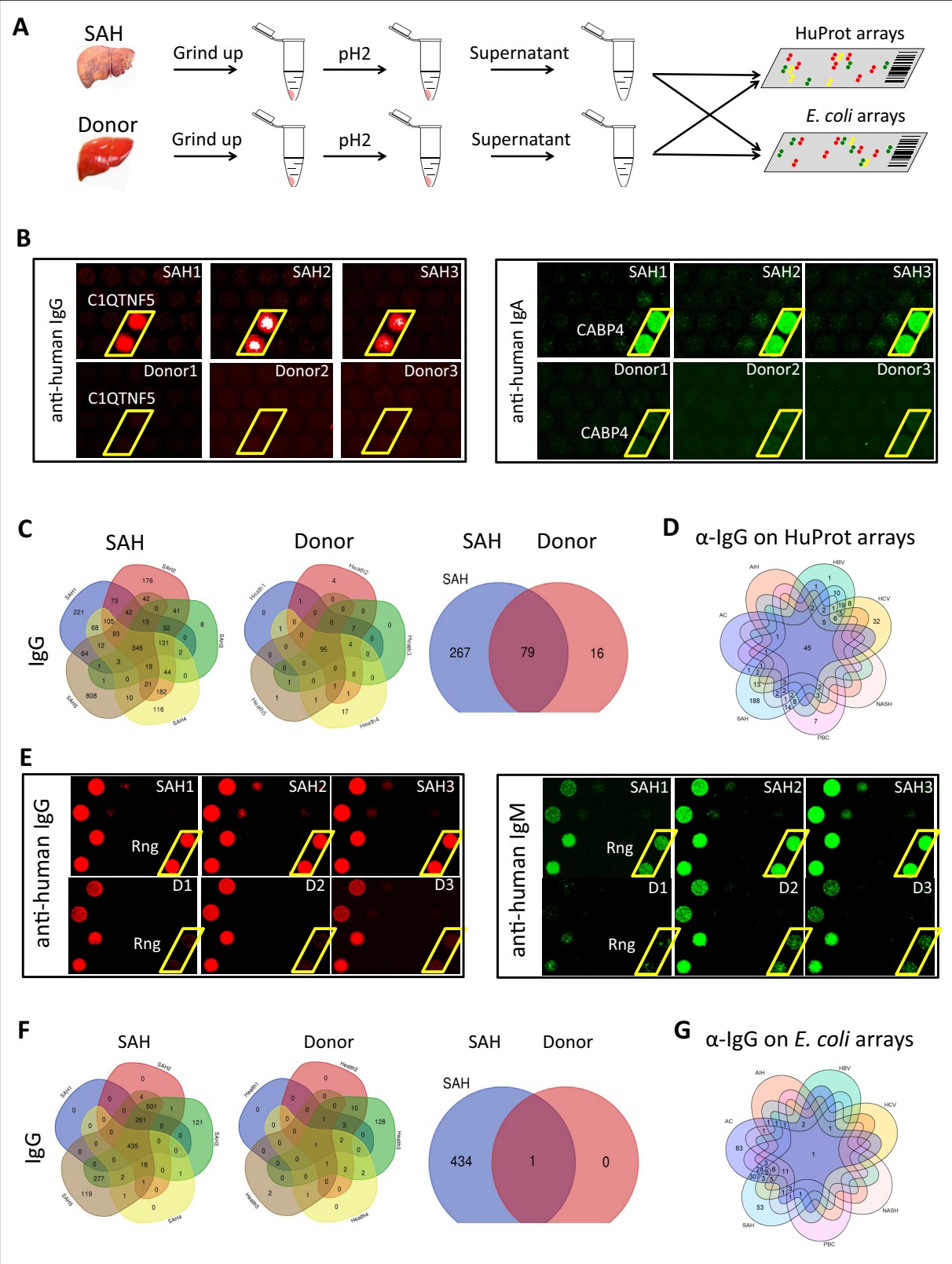

**Figure 4.** Human and *Escherichia coli* proteome arrays that identify a group of unique antibodies in the severe alcoholic hepatitis (SAH) liver recognize both human and bacterial antigens. (**A**) Each liver tissue piece was ground up and treated under low pH to release tissue-deposited Ig. After neutralization, the extracted Ig from each liver sample were separately probed to the HuProt or *E. coli* protein arrays, followed by incubation with the isotype-specific secondary antibodies to obtain the Ig isotype immune signatures of the same liver sample. (**B**) Representative images of HuProt

*Figure 4 continued on next page*

*Figure 4 continued*

arrays. (**C**) Venn diagram analysis to identify shared autoantigens of each liver disease. 346 autoantigens were shared by the IgG antibodies in all five SAH livers (left panel), 95 autoantigens were shared by the IgG antibodies in all five healthy donor (HD) livers (middle panel), and a large fraction (i.e., 267 proteins) of the SAH-shared IgG autoantigens was not found in the HD livers (right panel), suggesting existence of an SAH-specific autoimmune signature. (**D**) A seven-way Venn diagram analysis showed that 45 autoantigens were commonly recognized by IgG isotype autoantibodies in liver tissue homogenates extracted from different liver diseases, while 188 unique IgG autoantigens were recognized by tissue homogenates from the SAH livers. (**E**) Representative images of *E. coli* protein arrays. (**F**) Venn diagram analysis to identify shared *E. coli* antigens recognized by each liver disease. 435 *E. coli* antigens were commonly recognized by IgG antibodies in the five SAH livers (left panel), while only 1 *E. coli* antigen was commonly recognized by IgG antibodies in the five HD livers (middle panel). 434 out of 435 *E. coli* antigens were uniquely recognized by IgG antibodies in SAH livers but not HD livers (right panel). (**G**) A seven-way Venn diagram analysis showed that unique IgG bacterial antigens were only identified by using liver tissue homogenates from SAH or AC.

The online version of this article includes the following source data and figure supplement(s) for figure 4:

**Source data 1.** The number of unique autoantigens recognized by antibodies extracted from the diseased liver tissues (HuProt arrays).

**Source data 2.** The number of unique *E. coli* antigens recognized by antibodies extracted from the diseased liver tissues (*E. coli* proteome arrays).

**Figure supplement 1.** Numbers of autoantibodies shared by all five severe alcoholic hepatitis (SAH) livers.

**Figure supplement 2.** Venn diagram analysis of all liver samples.

**Figure supplement 3.** Unique autoantigens recognized by immunoglobulin (Ig) from diseased livers on the HuProt arrays.

**Figure supplement 4.** Venn diagram analysis of antigens identified on the *E. coli* proteome arrays.

**Figure supplement 5.** Unique bacterial antigens recognized by immunoglobulin (Ig) from diseased livers on the *E. coli* proteome arrays.

IgG antibodies in the five AC livers (*Figure 4—figure supplement 4*). The numbers of the commonly shared bacterial antigens identified by the IgG antibodies extracted from the other 5 liver diseases were much lower, ranging from 7 to 63 (*Figure 4—figure supplement 4*). In addition, the numbers of IgA-, IgM-, or IgE-recognized bacterial antigens by liver tissue homogenates from SAH or AC were much higher than that in other liver diseases, except PBC showing a higher number of IgM antigens than AC (*Figure 4—figure supplement 4*). This observation suggests that chronic or acute ALD is associated with specific anti-bacterial antibody signatures.

A large number of shared bacterial antigens were recognized by liver tissue homogenates from each liver disease. We found that 1, 6, 24, and 5 *E. coli* antigens were commonly recognized by IgG, IgA, IgM, and IgE isotype antibodies in liver tissue homogenates extracted from different liver diseases (*Figure 4F*, *Figure 4—figure supplement 5*). Unique IgG or IgA bacterial antigens were only identified by using liver tissue from SAH or AC (*Figure 4G*, *Figure 4—figure supplement 5*). More unique bacterial antigens were recognized by the SAH than AC IgA (110 vs 54), IgM (54 vs 1), and IgE (45 vs 1), but less so by the SAH IgG (53 vs 83) (*Figure 4—source data 2*). Notably, 69 unique IgM bacterial antigens were recognized by PBC liver tissue. Common anti-*E. coli* protein antibodies are present in diseased livers regardless of etiology, but unique anti-*E. coli* IgG and IgA antibodies exist only in alcoholic livers predominantly in SAH.

The prevalent anti-bacterial immunoactivity of the Ig in SAH livers suggested that Ig deposited in SAH livers might be derived from leaky gut and these anti-bacterial antibodies cross-react with human liver proteins. We used proteins from *E. coli* (strain K12) immobilized on magnetic beads to capture Ig pooled from the SAH livers (*Figure 5A*) (*E. coli*-captured Ig) which were then released and incubated on the HuProt arrays to determine whether these bacterium-recognizing antibodies could also cross-react with human proteins (examples shown in *Figure 5B*). At a stringent cutoff value (e.g., SD=10) 694, 796, 451, and 42 human proteins were reproducibly identified by *E. coli* protein-captured IgG, IgA, IgM, and IgE antibodies from the five SAH livers, respectively (*Figure 5C* and antibody-recognized protein sets in *Supplementary file 1*). Strikingly, many of these *E. coli* binding Ig recognized proteins (74.06%, 76.46%, 48.39%, and 49.41%, respectively) were also found to be the autoantigens recognized by Ig recovered directly from the five SAH livers. These results demonstrated that there exist a large number of cross-reacting antibodies that recognize both bacterial and human proteins in the SAH livers.

Gene ontology (GO) enrichment analysis of proteome arrays identified autoantigen-enriched unique common cellular components recognized by both Ig- and *E. coli*-captured Ig in SAH livers.

To determine if autoantigens recognized by Ig from diseased livers were specifically presented in cellular components, we performed GO enrichment analysis on antibody-recognized protein sets.

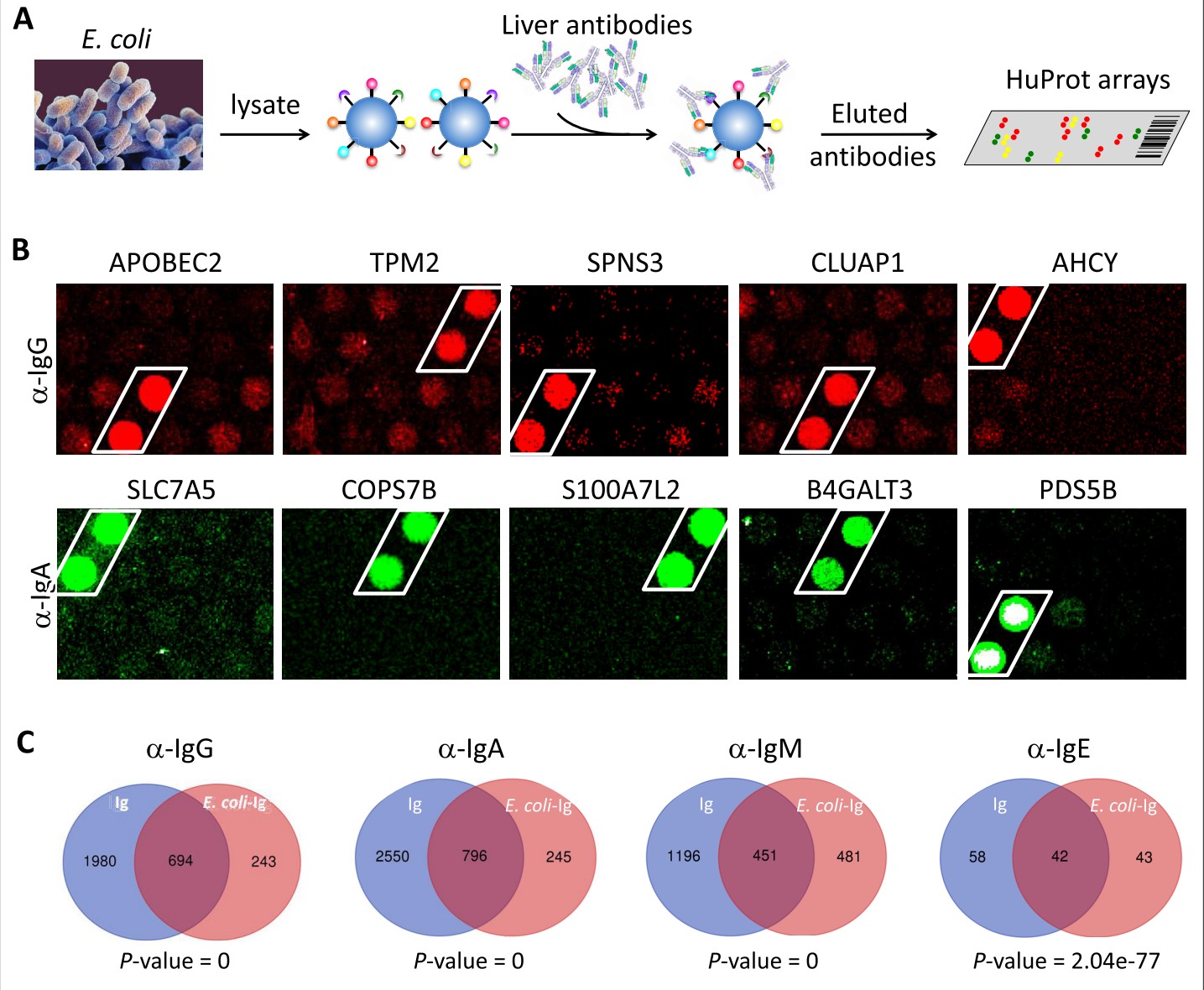

**Figure 5.** *E. coli* antigens-captured immunoglobulins (Ig) from severe alcoholic hepatitis (SAH) livers recognize human protein antigens. (**A**) To determine if anti-bacterial antibodies cross-react with human proteins in the liver, total proteins from *E. coli* (strain K12) were extracted and immobilized on NHS-activated magnetic beads to capture Ig pooled from the five SAH livers. *E. coli* protein-captured antibodies (*E. coli*-Ig) were then released and incubated on the HuProt arrays. (**B**) Representative images of *E. coli*-Ig on HuProt arrays. (**C**) 937, 1041, 932, and 85 human proteins were reproducibly identified by *E. coli* protein-captured IgG, IgA, IgM, and IgE antibodies from the five SAH livers. Venn diagram analysis showed many of these proteins (694/937, 796/1041, 451/932, and 42/95, respectively) were also found to be the autoantigens recognized by Ig recovered directly from the five SAH livers.

Proteins recognized by IgG antibodies from SAH livers were significantly over-represented in eight cellular components including cytosol, cytoplasm, nucleus, mitochondrion, focal adhesion extracellular exosome, mitochondrial intermembrane space, and ruffle membrane, while autoantigens recognized by IgA antibodies from SAH livers were over-represented in cytosol and cytoplasm (***Figure 6A***). These proteins are involved in a number of biological processes including nucleobase-containing small molecule interconversion, protein phosphorylation, signal transduction, regulation of cell proliferation, mitochondrial ribosome reassembly, and mitochondrion organization (***Figure 6B and C***). With the exception that cellular components were identified by IgM antibodies from PBC livers (***Figure 6— source data 1***), no cellular component was recognized by Ig from other diseased livers. Interestingly, *E. coli*-captured IgG and IgA from SAH livers recognized five out of eight cellular components which

**A**

#### Cellular components recognized by Ig from SAH livers (observed vs expected )

| Antigen enriched cellular component | Cytosol | Cytoplasm | Nucleus | Mitochondrion | Focal adhesion | Extracellular exosome | Mitochondrial intermembrane space | Ruffle membrane |
|---|---|---|---|---|---|---|---|---|
| IgG | P<0.0001 | P<0.0001 | P<0.0001 | P<0.05 | P<0.05 | P<0.05 | P<0.05 | P<0.05 |
| IgA | P<0.05 | P<0.05 | ns | ns | ns | ns | ns | ns |

**B**

#### IgG recognized unique autoantigens: biological process

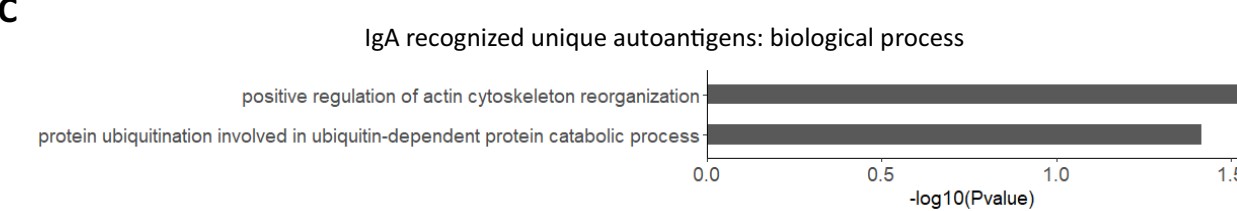

**C**

#### IgA recognized unique autoantigens: biological process

**D**

#### Common cellular components recognized by Ig and *E.coli* enriched Ig in SAH livers

| Antigen enriched cellular component | Cytosol | Cytoplasm | Nucleus | Mitochondrion | Focal adhesion |
|---|---|---|---|---|---|
| IgG | High | High | High | low | low |
| IgA | High | High | None | None | None |
| IgM | None | None | None | None | None |
| IgE | None | None | None | None | None |

High: p<0.0001; Low: p<0.05

**Figure 6.** Gene ontology (GO) enrichment analysis of proteome arrays identifies autoantigen-enriched common cellular components recognized by both immunoglobulin (Ig)- and *E. coli*-captured Ig in severe alcoholic hepatitis (SAH) livers. (**A**) Cellular components recognized by IgG and IgA antibodies in SAH livers. (**B–C**) Biological processes are involved by IgG autoantigens (**B**) and IgA autoantigens (**C**). (**D**) Common cellular components recognized by both Ig- and *E. coli* antigens-captured Ig in SAH livers.

*Figure 6 continued on next page*

*Figure 6 continued*

The online version of this article includes the following source data for figure 6:

**Source data 1.** Cellular components recognized by IgM and *E. coli* enriched IgM extracted from PBC liver tissues.

**Source data 2.** Cellular components recognized by IgG or IgA antibodies extracted from the diseased liver tissues (Observed vs. Expected).

were recognized by Ig directly extracted from SAH livers (*Figure 6D*). This was observed only in SAH livers (*Figure 6—source data 2*). These results further demonstrated the presence of unique cross-reacting anti-*E. coli* autoantibodies in the SAH livers.

## The infiltration of B and plasma cells in SAH livers is associated with increased Ig gene expression

Alcohol-derived leaky gut may promote translocation of gut bacterial products and Peyer's patches IgA-secreting plasma cells to the liver (*Moro-Sibilot et al., 2016*). To determine if the migration of bacteria and/or bacterial products from the bowel to liver occurred in SAH, we performed IHC staining for the gram-negative bacterial (*E. coli*) product livers. Both LPS and LTA were in liver tissue from SAH patients cf. controls (*Figure 7A*), especially in the inflammatory areas. The increase of LPS levels in SAH liver tissues was confirmed by using Pierce Chromogenic Endotoxin Quant Kit (*Figure 7B*). IHC staining for CD20+ and CD138+ cells revealed that most of these cells were localized in the inflammatory areas of SAH livers, with none in HD livers (*Figure 7C*). These results suggest that increased gut bacterial antigens in the SAH liver are associated with increased B cells and plasma cells.

Based on our findings that Ig extracted from SAH livers but not present in the serum exhibited hepatocyte killing efficacy (*Figure 2G*), we suspected that Ig antibodies deposited in hepatocytes might not be produced systemically but by plasma cells infiltrating the liver. To determine if infiltration of B and plasma cells in the SAH liver is associated with expression of genes coding for certain Ig or fragments of Ig, we analyzed RNA-seq data generated from SAH and HD livers (GEO GSE143318). Specifically, we examined whether genes coding for Ig or fragments of specific Ig are upregulated in SAH liver samples relative to donor liver samples. We identified 175 genes coding for Ig or fragments of Ig and compared their expression levels between SAH and control samples. Interestingly, 87 identified genes representing Ig fragments showed a robust and stable expression in SAH livers but not in HD livers (*Figure 7D*). These results suggest that infiltrating B and plasma cells secrete Ig that not only have anti-intestinal antigen activity but also cross-react with hepatocyte antigens.

## Discussion

Using explanted livers from SAH patients, we discovered massive antibody deposition in ballooned hepatocytes which is associated with complement activation. Antibodies from the SAH liver but not patient serum exhibited hepatocyte killing efficacy in vitro. Unique antibodies found in the SAH liver not only recognize bacterial (*E. coli*) antigens but also cross-react with a large number of human antigens. Our data support the hypothesis that anti-bacterial antibodies and/or plasma cells originating from gut translocate to the liver due to gut leakiness caused by excessive alcohol drinking and that these hepatic plasma cells produce antibodies which not only recognize intestinal bacterial antigens but also cross-react with antigens of human hepatocytes (*Figure 6E*). Antibody deposition, with complement activation as well as immune cell activation, may result in acute liver failure due to antibody-mediated inflammation.

Leaky gut, also known as increased intestinal permeability, is a digestive condition in which bacteria and toxins are able to pass from the gastrointestinal tract to extraintestinal sites and facilitate an increased ingress of inflammatory cytokines and endotoxin to the liver (*Bajaj, 2019*). Unique anti-*E. coli* IgG or IgA antibodies are identified in alcoholic livers, but not other liver diseases (*Figure 4—source data 2*), suggesting the impact of excessive alcohol drinking on translocation of anti-bacterial antibodies to the liver. Interestingly, few anti-bacterial antibodies in AC livers cross-reacted with human antigens (*Figure 4—figure supplement 4*), while a large number of anti-bacterial antibodies in SAH livers also recognized human antigens. The differing gut microbiota between actively drinking patients with cirrhosis and those with alcoholic hepatitis (*Llopis et al., 2016*; *Llorente et al., 2017*; *Duan et al., 2019*) supports the notion that alcohol-induced changes of gut microbiota composition

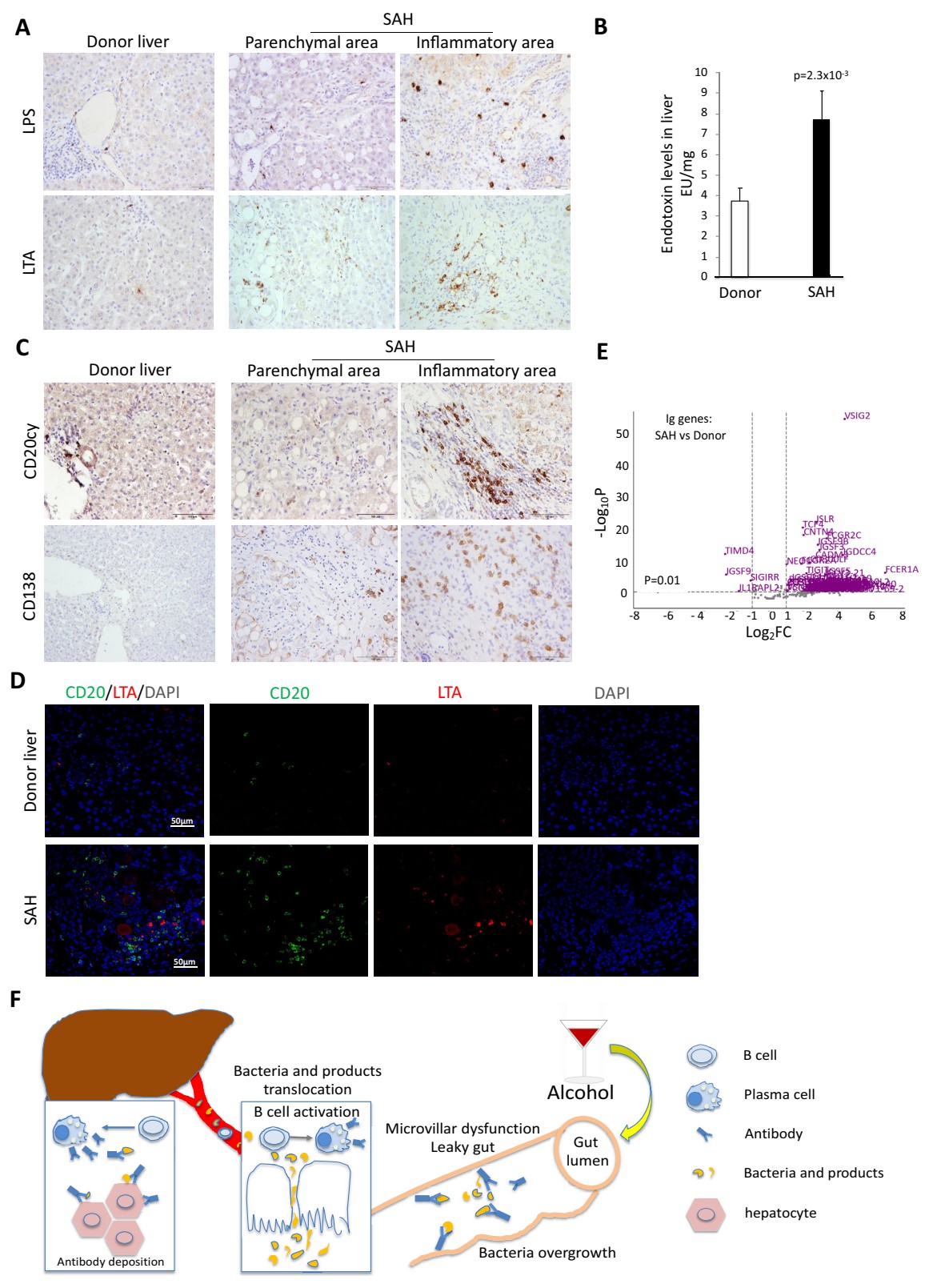

**Figure 7.** Increased bacterial antigens, B and plasma cell infiltration, and immunoglobulin gene expression in severe alcoholic hepatitis (SAH) livers. (**A**) Immunohistochemistry staining for the gram-negative bacterial product lipopolysaccharides (LPS) and gram-positive bacteria antigen lipoteichoic acid (LTA) in SAH livers. (**B**) LPS levels in SAH liver tissues were quantified by using Pierce Chromogenic Endotoxin Quant Kit (n=7/group). (**C**) Immunohistochemistry staining for CD20+ B cells and CD138+ plasma cells in SAH livers. Representative images from 20 SAH or 10 healthy donor

*Figure 7 continued*

(HD) livers (**A–C**). (**D**) Double immunofluorescence staining with CD20 (green) and LTA (red) showed colocalization of B cells and bacteria antigens in the inflammatory areas. Representative tissue sections from six samples per group. (**E**) RNA-sequencing analysis identified differentially expressed genes representing immunoglobulin fragments in SAH livers. Purple dots represented 91 differentially expressed immunoglobulin genes (log$_2$FC >1 and adjusted p-value <0.01) in the comparison of SAH patients between normal donors. Among them, 87 were upregulated and 4 downregulated. Gray dots were the remaining immunoglobulin genes that did not meet the thresholds. n=7 per group. (**F**) Schematic representation of mechanism of cross-reacting anti-bacterial autoantibody-mediated hepatocyte damage in SAH.

may contribute to the development of anti-bacterial antibodies that cross-reacted with human antigens. SAH is an acute-on-chronic liver failure. Cross-reacting anti-bacterial autoantibodies translocated to the liver may determine the progress from chronic liver disease toward acute hepatitis.

Coincidentally, we found a significant number of unique IgA autoantibodies in HBV and unique IgG autoantibodies in HCV livers (*Figure 4—source data 1*), suggesting these IgA or IgG autoantibodies deposited in the liver might be derived from systemic autoimmune process which is ongoing in HBV- or HCV-infected patients (*Lin et al., 2016*; *Gonzàlez-Quintela et al., 2003*). Previous studies have clearly established the co-occurrence of certain autoantibodies in patients with chronic HBV or HCV infection and the autoantibody positivity was common in HBV and HCV patients. A strong association between increasing serum IgA level and disease progressing in patients with chronic HBV infection has been reported (*Lin et al., 2016*). Similarly, serum IgG was increased in patients with chronic hepatitis C with respect to both alcoholics and healthy controls (*Gonzàlez-Quintela et al., 2003*). No unique anti-*E. coli* protein IgA or IgG antibody was identified in HBV or HCV livers, indicating autoantibody production was related to the virus rather than bacterial infection.

The presence of cross-reacting IgM antibodies against *E. coli* and human antigens in the PBC livers is worth noting. Current data suggest that PBC is an autoimmune disease (*Bogdanos et al., 2004*; *Kikuchi et al., 2005*) and an infectious etiology as trigger for development of PBC has been postulated. Antibodies reacting against the mitochondrial human pyruvate dehydrogenase complex which are the serologic hallmark of PBC cross-react with the *E. coli* pyruvate dehydrogenase complex, implicating *E. coli* infection in the pathogenesis of PBC (*Wang et al., 2014*). Interestingly, patients with PBC had a much higher number of inducible IgM-producing B cells in peripheral blood and each B cell produced a greater quantity of IgM protein compared with control (*Kikuchi et al., 2005*). A large number of unique human and *E. coli* antigens were recognized specifically by IgM from the PBC livers, supporting the theory of bacterial infection-related IgM production in pathogenesis of PBC.

Our findings suggest that SAH may be an antibody-mediated disease due to intrahepatic etiology. Unlike antibody-mediated autoimmune disease where systemic B cells produce autoantibodies against self-antigens and unlike the fact that autoimmune diseases can be transferred from an affected patient to a normal individual by the transfer of patient-derived serum (or Ig), serum from SAH patients did not show hepatocyte killing efficacy in vitro. Importantly, no antibody-mediated rejection was observed in SAH patients following liver transplantation (*Weeks et al., 2018*). It is perhaps more likely that gut-derived plasma cells were resident in SAH livers, and these plasma cells produce anti-bacteria antibodies which strongly cross-react with hepatocyte antigens. The diseased liver absorbs these gut-derived and locally produced antibodies without releasing them to the circulation. With the functional deposition of antibodies in the SAH liver (*Figure 2*), activation of complement through the classical pathway which has been reported previously (*Shen et al., 2014*) leads to ballooning degeneration of hepatocytes which is the predominant mode of injury in alcoholic hepatitis and untreatable inflammation. When antibody is deposited, it may also impair a number of important biological process in hepatocytes (*Figure 6*). This pathophysiology implies severe damage from short-lived leaky gut due to a transient interval of high alcohol intake with ongoing residual continued damage due to the antibody production that continues only within the liver to be replaced. Therefore, it is reassuring that the new transplanted liver will not be damaged because the pathogenesis, involving the consequences of leaky gut pathology, will end quickly as long as the recipient remains abstinent, by which the new liver is not threatened by the SAH process.

The main limitations of the present study are the limited sample size, the lack of whole gut bacterial-proteome microarrays, and the lack of animal models of SAH. Nevertheless, identifying the presence of anti-bacterial autoantibodies in SAH livers provides a new insight into pathogenesis of SAH. Future studies utilizing needle biopsy samples from mild and moderate alcoholic hepatitis patients could

assess infiltration of B and plasma cells and a likely correlation between antibody deposition and severity of hepatitis. For example, infiltration of plasma cells and antibody deposition may predict the progression toward SAH.

This may be a new therapeutic strategy in alcoholic hepatitis patients. First, preventing gut leakiness caused by alcohol drinking may prevent SAH. Second, eliminating antibody-producing B/plasma cells in the liver by using anti-CD20 may serve. Finally, a trial using complement inhibitors such as eculizumab may be worth considering.

## Methods

### Collection of liver tissue samples

Explanted liver tissues and blood were collected from patients with SAH or AC who were referred for liver transplantation at Johns Hopkins Hospital after informed consent to study, share, and publish the research data derived from their specimens (*Weeks et al., 2018*). Explanted liver tissues from patients with other liver diseases were obtained from the Liver Tissue Procurement and Distribution System at the University of Minnesota, which was funded by NIH Contract# HHSN276201200017C. All studies were approved by the Johns Hopkins Medicine Institutional Review Boards (IRB00107893 and IRB00154881).

### Immunohistochemistry

Cut sections were prepared from formalin-fixed paraffin-embedded liver tissues for staining with IgG (ab200699), IgA (ab200699 or GTX20770), C4d (ab167093), C3d (ab136916), CD20 (Dako, Santa Clara, CA, USA), CD138 (Abcam, Cambridge, MA, USA), *E. coli* LPS (Abcam, Cambridge, MA, USA), LTA (Thermo Fisher, Waltham, MA, USA). Vectastain Elite ABC Staining Kit and DAB Peroxidase Substrate Kit (Vector Laboratories, Burlingame, CA, USA) were used to visualize the staining according to the manufacturer's instructions. Diaminobenzidine tetrahydrochloride and blue alkaline phosphatase (Vector Laboratories) were used as brown and blue chromogen and hematoxylin as nuclear counterstaining. Echo Revolve microscope (Echo Laboratories Inc) was used for taking image pictures.

### ELISA

Liver protein lysates used for this assay contained similar concentrations of protein. Each SAH and donor liver sample were added on a precoated ELISA plate (Thermo Fisher Scientific, Waltham, MA, USA) to determine the total IgG (BMS2091), IgA (BMS2096), IgM (BMS2098), and IgG subclasses (991000).

### Isolation of PBMCs

PBMCs were isolated from heparinized peripheral blood samples from healthy volunteers using Ficoll-Paque plus density gradient medium.

### ADCC assay

ADCC was determined by a calcein-acetyoxymethyl release assay (calcein-AM, C3100MP, Thermo Fisher Scientific). Calcein-AM-labeled primary human hepatocytes were cultured with Ig from SAH livers, patient serum, or human IgG isotype control in a 96-well plate at a density of $1\times10^4$ cells per well in triplicate and PBMCs were added as effector cells at an effector: target cell ratio of 5:1 respectively. Antibody-independent cell-mediated cytotoxicity (AICC) was measured in wells containing target and effector cells without the addition of AH or control IgG antibodies. The following formula was used to calculate ADCC: % ADCC = 100 × (mean experimental release – mean AICC) ÷ (mean maximum release – mean spontaneous release).

### Western blot analysis

Fifty milligram liver tissues from SAH or the controls were homogenized in the lysis buffer (Cell Signaling Technology, MA, USA). The total protein concentrations of each liver samples were determined using a standard curve generated with BSA at different known concentration using the Quick Start Bradford Protein Assay (Bio-Rad, USA). On the basis of the measure protein concentrations, 25 μg of total proteins of each liver sample were boiled in NuPAGE LDS Sample Buffer

(Thermo Fisher, MA, USA) and subjected to electrophoresis in a 4–12% Bis-Tris gradient PEG gel (Thermo Fisher, MA, USA). All samples were tested in several parallel gels. One gel was stained with SimplyBlue SafeStain according to the product manual (Thermo Fisher, MA, USA), and the other one was subjected to the western blot assay using the Trans-Blot Turbo RTA Midi 0.45 μm LF PVDF Transfer Kit (Bio-Rad Laboratories, CA, USA). After transferring the total proteins to the PVDF membrane, the membranes were incubated with the IRDye-labeled human Ig antibodies specific to recognizing IgG, IgM, IgA, and IgE, or mouse monoclonal antibodies against human C3d (Bio-Rad Laboratories, CA, USA) and C4d (Santa Cruz, CA, USA) respectively, followed by probing with Alexa 647-labeled Goat anti-Mouse IgG (H+L) (Thermo Fisher, MA, USA). After scanning with Odyssey CLx Imaging System, the signals were calculated by the corresponding software and then analyzed by Excel.

## Multiplex immunofluorescence staining

Sequential multiplex immunofluorescence staining on formalin-fixed, paraffin-embedded liver sections was performed, as previously described(*Guillot et al., 2020*). Images were acquired on a Zeiss LSM 900 confocal microscope. Acquired images were processed and analyzed using FIJI (*Schindelin et al., 2012*). The following antibodies were used: HepPar1 (Catalog NBP2-45272, Novus), CD68 (Catalog M0876, Dako), CD32 (Catalog 53151, Cell Signaling Technology), IgG (109-005-088, Jackson ImmunoResearch), IgA (Catalog ab124716, Abcam), β-catenin (Catalog 610154, BD Biosciences), C4d (Catalog BI-RC4D, BIOMEDICA).

## Antibody extraction and protein microarray analysis

The liver tissues were subjected to a Dounce homogenizer in a lysis buffer (0.1 M glycine pH 2.0, 150 mM NaCl) to elute the binding antibodies (Ig). After a 5-min spindown with 20,000 × *g* at 4°C, the supernatants were transferred to new 15 ml tubes and neutralized with 1 M Tris-base buffer to pH 7.0 immediately. Then, the antibody enrichment was performed using Protein L-coupled magnetic beads (Thermo Fisher, MA, USA) according to the manual. The antibodies extracted from 4.8 g liver tissue pieces were subjected to the protein microarray assays using human proteome microarray HuProt array and *E. coli* strain K-12 bacterial proteome microarray respectively to screen their corresponding antigens (*Hu et al., 2017*; *Chen et al., 2009*). Data analysis including the criteria for positive hits was performed as before (*Hu et al., 2017*; *Chen et al., 2009*).

## Identification of cross-reactive antibodies against human and bacterial proteins

The total lysates of *E. coli* were immobilized on the magnetic beads using the kit of Pierce NHS-Activated Magnetic Beads according to the product manual (Thermo Fisher, MA, USA). Then, these *E. coli* protein magnetic beads were incubated with the extracted total autoantibodies from liver samples to capture the active antibodies. After eluting with glycine pH 2.0, the eluted antibodies were neutralized with 1 M Tris-base buffer to pH 7.0 immediately. Finally, these antibodies were subjected to the human proteome microarray.

## RNA-sequencing and data processing

The total RNA was isolated using QIAGEN RNeasy kit. After the RNA quality was assessed by capillary electrophoresis (Bioanalyzer), cDNA libraries were prepared using TruSeq RNA Library Prep Kit and sequenced with an Illumina NextSeq500. Base-calling and fastq conversion was performed using RTA (2.4.11) and Bcl2fastq (2.18.0.12), respectively. Raw sequencing files were uploaded to the NIH GEO database.

Adaptor sequences were trimmed from the raw reads using Cutadapt (*Martin, 2011*). Trimmed reads were then mapped to reference genome GRCh38 using STAR aligner with default parameters (*Dobin et al., 2013*). The number of counts per gene was estimated using the 'quantMode' command in STAR. Batch effect was corrected using Combat seq. Differentially expressed genes (DEGs) were then identified using DESeq2 (*Love et al., 2014*). Genes with adjusted p<0.01 and $\log_2$ fold change >1 were chosen as DEGs.

## GO analysis

DAVID (*Huang et al., 2009*) was used to conducted GO analysis to find out enriched GO terms (cellular components and biological process). All enriched terms were chosen with a threshold p-value of 0.05.

## Acknowledgements

The authors thank patients and liver donors, their families and surrogates and medical personnel. We thank the anesthesiologists, nurses, and transplantation fellows at the Johns Hopkins Hospital who assisted in collecting samples. In addition, we thank clinical pathology core at Johns Hopkins Hospital for performing IHC staining and scanning the slides. Funding. This work was funded by NIH grants R24AA02517 (ZS), P50AA027054 (Project 1 [AMC], Project 3 [HZ and ZS]), and ZIAAA000368 (BG). The funders had no role in study design, data collection, and analysis, decision to publish or preparation of the manuscript.

## Additional information

### Funding

| Funder | Grant reference number | Author |
|---|---|---|
| National Institute on Alcohol Abuse and Alcoholism | R24AA02517 | Zhaoli Sun |
| National Institute on Alcohol Abuse and Alcoholism | P50AA027054 | Heng Zhu Zhaoli Sun |
| National Institute on Alcohol Abuse and Alcoholism | ZIAAA000368 | Bin Gao |

The funders had no role in study design, data collection and interpretation, or the decision to submit the work for publication.

### Author contributions

Ali Reza Ahmadi, Conceptualization, Data curation, Formal analysis, Investigation, Methodology, Writing – original draft, Writing – review and editing; Guang Song, Data curation, Formal analysis, Investigation, Methodology, Writing – review and editing; Tianshun Gao, Xiaomei Han, Ming-Wen Hu, Brandon Peiffer, Robert Anders, Data curation, Formal analysis, Methodology, Writing – review and editing; Jing Ma, Formal analysis, Investigation, Methodology, Writing – review and editing; Andrew M Cameron, Resources, Investigation, Writing – review and editing, Funding acquisition; Russell N Wesson, Shane Ottmann, Elizabeth King, Ahmet Gurakar, Resources, Investigation, Writing – review and editing; Benjamin Philosophe, Chien-Sheng Chen, Resources, Writing – review and editing; Le Qi, Formal analysis, Methodology, Writing – review and editing; James Burdick, Writing – review and editing; Zhanxiang Zhou, Hongkun Lu, Investigation, Methodology, Writing – review and editing; Dechun Feng, Investigation, Methodology, Writing – review and editing, Formal analysis; Jiang Qian, Data curation, Software, Formal analysis, Supervision, Validation, Investigation, Methodology; Bin Gao, Writing – review and editing, Funding acquisition, Investigation, Methodology, Supervision; Heng Zhu, Data curation, Formal analysis, Supervision, Funding acquisition, Validation, Investigation, Methodology, Writing – review and editing; Zhaoli Sun, Conceptualization, Data curation, Supervision, Funding acquisition, Writing – original draft, Project administration, Writing – review and editing, Formal analysis, Investigation, Methodology, Resources

### Author ORCIDs

Ali Reza Ahmadi (ID) http://orcid.org/0000-0001-8140-2690
Guang Song (ID) http://orcid.org/0000-0002-7630-716X
Tianshun Gao (ID) http://orcid.org/0000-0002-0466-9081
Brandon Peiffer (ID) http://orcid.org/0000-0002-6783-2923

Chien-Sheng Chen http://orcid.org/0000-0002-2372-324X
Jiang Qian http://orcid.org/0000-0002-0476-3596
Heng Zhu https://orcid.org/0000-0002-8426-2889
Zhaoli Sun http://orcid.org/0000-0002-4272-8046

## Ethics

Explanted liver tissues and blood were collected from patients with SAH or AC who were referred for liver transplantation at Johns Hopkins hospital after informed consent to study, share and publish the research data derived from their specimens. These activities were approved by the Johns Hopkins Medicine Institutional Review Boards (IRB00107893 & IRB00154881).

Reviewer #1 (Public Review): https://doi.org/10.7554/eLife.86678.2.sa1
Reviewer #2 (Public Review): https://doi.org/10.7554/eLife.86678.2.sa2
Author Response https://doi.org/10.7554/eLife.86678.2.sa3

# Additional files

## Supplementary files

• Supplementary file 1. Human protein (autoantigen) sets recognized by both immunoglobulins (Ig) and *E. coli*-captured Ig from severe alcoholic hepatitis (SAH) livers.

• MDAR checklist

## Data availability

External data requests can be directed to the corresponding authors, who will respond within 1 week and help facilitate the request. Raw sequencing files were uploaded to the NIH GEO database under accession number GSE143318.

The following dataset was generated:

| Author(s) | Year | Dataset title | Dataset URL | Database and Identifier |
|---|---|---|---|---|
| Sun Z, Qian J | 2020 | RNAseq analysis highlights significant transcriptional changes within the livers of patients with alcoholic hepatitis | https://www.ncbi.nlm.nih.gov/geo/query/acc.cgi?acc=GSE143318 | NCBI Gene Expression Omnibus, GSE143318 |

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
