## [Editor Report · eLife assessment]

This **important** study tested the hypothesis that liver-derived but not serum-derived antibodies that are cross-reactive to E.coli and to host proteins can play a role in the hepatic damage found in severe alcoholic hepatitis (SAH). Using a **solid** methodology that includes state-of-the-art microscopy, proteome arrays, and gene ontology assays, it provides strong evidence that liver-derived IgG and IgA with cytotoxic properties and reactivity to both gut-derived *E. coli* and autoantigens accumulated in hepatocytes of SAH patients but not of healthy controls. The study would benefit from a broader analysis of gut microbiota proteome and further characterization of B cells infiltrating the liver tissue including their numbers/field and their origin (infiltrating versus resident cells). The work opens new avenues of understanding for the pathogenesis of severe alcoholic hepatitis and is of great interest to researchers and clinicians in the field.

---

## [Referee Report · Reviewer #1 (Public Review)]

Sun and colleagues investigated the cross-reactive antibodies between *E. coli* and the host in severe alcoholic hepatitis (SAH). The study found that IgA and IgG were deposited in the liver of SAH patients. Complements C3d and C4d were also deposited in the SAH patient's liver. Moreover, they found that the Ig accumulated in the SAH liver, but not in the SAH serum, induced hepatocyte killing, suggesting that liver Ig is important. Then, they found that these Ig can recognize both human and *E. coli* antigens. Very interestingly, SAH-derived Ig shows cross-reactivity to both human and *E. coli* antigens, suggesting *E, coli*-primed Ig in SAH may damage hepatocytes through host antigen recognition. These Ig are not observed in alcoholic cirrhosis patients. The liver RNA-seq data suggested that Ig was also produced in the liver, not only gut-derived Ig. This is a very interesting study showing the novel mechanism of SAH mediated by the Ig with the cross-reactivity with bacteria and host antigens, which is not observed in AC patients. Overall, the study design is reasonable and the data are consistent to support their central hypothesis. There are a few comments.

Specific comments:

1. Figures 1 and 2 show Ig deposition in the liver (it seems on hepatocytes). Not only Ig reaction to the specific antigen but also non-specific Fc receptor-mediated binding to hepatocytes could be contributed.

2. Similarly, in Figure 2G Ig-mediated hepatocyte killing, Fc receptor-mediated hepatocyte killing may be involved.

3. The study examined the possibility of liver resident B cell and plasma cell-mediated Ig. As the authors mentioned in the discussion, B cells may be translocated from the intestine to the liver. Or the resident B cells (not from the gut) are also involved.

---

## [Referee Report · Reviewer #2 (Public Review)]

In this paper, Ahmadi et al demonstrated that antibodies produced locally in the liver by infiltrating B cells can enhance liver damage caused by fat accumulation. The main finding is that human samples extracted from severe alcoholic hepatitis showed antibody accumulation that may be related to an enhanced immune response to self-antigens, which could ultimately fuel liver damage - which was already present due to alcohol consumption. Their data are corroborated by arrays and gene ontology assays, and I strongly believe that these data could add to the future options we have to treat patients.

---

## [Author Response]

**Reviewer #1 (Public Review):**
Sun and colleagues investigated the cross-reactive antibodies between *E. coli* and the host in severe alcoholic hepatitis (SAH). The study found that IgA and IgG were deposited in the liver of SAH patients. Complements C3d and C4d were also deposited in the SAH patient's liver. Moreover, they found that the Ig accumulated in the SAH liver, but not in the SAH serum, induced hepatocyte killing, suggesting that liver Ig is important. Then, they found that these Ig can recognize both human and *E. coli* antigens. Very interestingly, SAH-derived Ig shows cross-reactivity to both human and *E. coli* antigens, suggesting *E. coli*-primed Ig in SAH may damage hepatocytes through host antigen recognition. These Ig are not observed in alcoholic cirrhosis patients. The liver RNA-seq data suggested that Ig was also produced in the liver, not only gut-derived Ig. This is a very interesting study showing the novel mechanism of SAH mediated by the Ig with the cross-reactivity with bacteria and host antigens, which is not observed in AC patients. Overall, the study design is reasonable and the data are consistent to support their central hypothesis. There are a few comments.

We thank the Reviewer for his/her positive comments on our manuscript!

Specific comments:1. Figures 1 and 2 show Ig deposition in the liver (it seems on hepatocytes). Not only Ig reaction to the specific antigen but also non-specific Fc receptor-mediated binding to hepatocytes could have contributed.1. Similarly, in Figure 2G Ig-mediated hepatocyte killing, Fc receptor-mediated hepatocyte killing may be involved.

Anti-IgG antibody (ab200699) recognizes a protein of 75 kDa, identified as gamma heavy chain of human immunoglobulins. It is possible that non-specific Fc receptor-mediated binding to hepatocytes in the SAH liver can also be recognized by this anti-IgG antibody staining.

However, no IgG or IgA deposition in the healthy donor livers was identified by anti-IgG or IgA staining. These results suggest that there was no antigen specific or Fc receptor-mediated binding to healthy hepatocytes.

In the ADCC assay, hepatocytes isolated from healthy donor livers were used as the target cells. Immune cell (NK) mediated ADCC is mainly triggered by IgG (binding to antigens of hepatocytes) through the interaction between IgG Fc fragment and Fc-receptors (FcγRs) of NK cells. If IgG deposition in the SAH liver were mainly due to non-specific Fc receptor-mediated binding to hepatocytes, we would expect IgG binding to FcγRs of hepatocytes and no activation of NK cells. Ig-mediated hepatocyte killing (Figure 2G) indicates the Ig (from SAH liver) reaction to the specific antigens.

1. The study examined the possibility of liver resident B cell and plasma cell-mediated Ig. As the authors mentioned in the discussion, B cells may be translocated from the intestine to the liver. Or the resident B cells (not from the gut) are also involved.

We agree with the Reviewer at this point. The resident B cells may be also involved in the Ig production.

**Reviewer #2 (Public Review):**
In this paper, Ahmadi et al demonstrated that antibodies produced locally in the liver by infiltrating B cells can enhance liver damage caused by fat accumulation. The main finding is that human samples extracted from severe alcoholic hepatitis showed antibody accumulation that may be related to an enhanced immune response to self-antigens, which could ultimately fuel liver damage - which was already present due to alcohol consumption. Their data are corroborated by arrays and gene ontology assays, and I strongly believe that these data could add to the future options we have to treat patients.

We thank the Reviewer for his/her positive comments on our manuscript!